# Serum 5-Hydroxyindoleacetic Acid and Ratio of 5-Hydroxyindoleacetic Acid to Serotonin as Metabolomics Indicators for Acute Oxidative Stress and Inflammation in Vancomycin-Associated Acute Kidney Injury

**DOI:** 10.3390/antiox10060895

**Published:** 2021-06-02

**Authors:** Hyun-Seung Lee, Sang-Mi Kim, Ja-Hyun Jang, Hyung-Doo Park, Soo-Youn Lee

**Affiliations:** 1Department of Laboratory Medicine and Genetics, Samsung Medical Center, Sungkyunkwan University School of Medicine, Seoul 06351, Korea; hyunseung1011.lee@samsung.com (H.-S.L.); jeehee0520@gmail.com (S.-M.K.); jahyun.jang@samsung.com (J.-H.J.); nayadoo@hanmail.net (H.-D.P.); 2Department of Clinical Pharmacology & Therapeutics, Samsung Medical Center, Sungkyunkwan University School of Medicine, Seoul 06351, Korea; 3Department of Health Science and Technology, Samsung Advanced Institute of Health Science and Technology, Sungkyunkwan University, Seoul 06351, Korea

**Keywords:** tryptophan, serotonin, 5-hydroxyindoleacetic acid, vancomycin, nephrotoxicity, oxidative stress, inflammation

## Abstract

The incidence of vancomycin-associated acute kidney injury (VAKI) varies from 5–43%, and early detection of VAKI is important in deciding whether to discontinue nephrotoxic agents. Oxidative stress is the main mechanism of VAKI, and serotonin (5-HT) and its metabolite 5-hydroxyindoleacetic acid (5-HIAA) have been examined with respect to their involvement in ischemia/reperfusion damage in experimental animal models. In the current study, we assessed 5-HT and 5-HIAA as novel biomarkers for detecting VAKI in patients who have infections or compromised renal function, using a mass spectrometry–based metabolomics approach. We conducted amino acid profiling analysis and measurements of 5-HT and 5-HIAA using serum from subjects with VAKI (*n* = 28) and non-VAKI control subjects (*n* = 69), consisting of the infection subgroup (*n* = 23), CKD subgroup (*n* = 23), and healthy controls (HCs, *n* = 23). 5-HT was significantly lower in the VAKI group than in the non-VAKI groups, and the concentration of 5-HIAA and the ratio of 5-HIAA to 5-HT (5-HIAA/5-HT) showed higher values in the VAKI group. The infection subgroup presented a significantly greater 5-HIAA/5-HT ratio compared with the HC subgroup. Our study revealed that increased 5-HIAA/5-HT ratio has the potential to act as a VAKI surrogate marker, reflecting acute oxidative stress and inflammation.

## 1. Introduction

Acute kidney injury (AKI) is a serious complication with high associated rates of mortality and morbidity among hospitalized patients [1]. Sepsis is the leading cause of AKI, and 45% to 70% of all AKI cases can be linked to sepsis [2]. For sepsis management, it is often necessary to use antibiotic agents with the potential to cause AKI, although nephrotoxic drugs were contributing factors in 16% to 25% of cases of severe AKI [3]. Vancomycin, one nephrotoxic drug [4,5,6], is a widely used glycopeptide antimicrobial against Gram-positive organisms, such as methicillin-resistant Staphylococcus aureus (MRSA), in clinical practice [6,7]. As a subtype of AKI, vancomycin-associated AKI (VAKI) is characterized by dose-dependent acute tubular necrosis, which is associated with oxidative stress in proximal tubule cells [5]. VAKI is correlated closely with elderly, compromised renal function, concomitant nephrotoxic therapy, longer duration of vancomycin administration, high trough level of vancomycin, and critically ill status in the intensive care unit [4,5,6]. The incidence of VAKI has been reported to range from 5% to 43% [6], and therapeutic drug monitoring and customized dosing of vancomycin have been studied for the prevention of VAKI in a high-risk population [6].

The rapid and accurate diagnosis of VAKI is important for decision-making about MRSA treatment [8]. Earlier detection of VAKI with a kidney-specific biomarker can result in discontinuation of nephrotoxic agents and the earlier initiation of alternate therapeutics [8,9]. An increase in serum creatinine concentration (SCr) and a reduction in urine output are the current diagnostic markers for VAKI [5,6,10]; however, these traditional markers of kidney function might not be sensitive to kidney damage, and their changes lag several days behind actual changes in glomerular filtration rate (GFR) [11]. Recent studies have shown that novel biomarkers of kidney damage, including kidney injury molecule–1, neutrophil gelatinase-associated lipocalin, liver-type fatty acid–binding protein, and interleukin-18, increase before SCr [12,13,14,15,16]. However, it is unclear whether these biomarkers could reveal the different AKI-inducing mechanisms and conditions, such as toxicity, nephron obstruction, ischemia, and inflammation [10,11]. A viable novel marker for VAKI must be suitable for detecting kidney damage in septic patients with or without compromised renal function [17].

Recent studies have attempted the prediction of drug-induced nephrotoxicity using a metabolomics approach [18,19]. This approach could be used to identify novel kidney biomarkers and might elucidate biological mechanisms of disease [20]. Zhang et al. reported that branched-chain amino acids could be predictors for individual differences of cisplatin-induced nephrotoxicity [21]. Oxidative stress is the main mechanism of VAKI, similar to cisplatin-induced nephrotoxicity [6,10]. Therefore, we hypothesized that changes in amino acid concentrations in the blood could be distinguished between patients with VAKI and those with infection or compromised renal function. Furthermore, specific metabolites of amino acids and their ratios have the potential to act as novel biomarkers of VAKI.

In this study, we conducted an exploratory metabolome profiling analysis and amino acids profiling analysis from clinical residual samples obtained from VAKI patients and non-VAKI patients, including those with infection and chronic kidney disease (CKD). Using this approach, we aimed to analyze alterations in the metabolomics profile in acute and chronic oxidative stress conditions and to discover a novel biomarker of VAKI.

## 2. Materials and Methods

### 2.1. Study Cohort

This study enrolled a total of 97 subjects. A total of 3370 samples with supratherapeutic level of vancomycin (vancomycin trough concentration (C_min_) > 20 μg/mL) [6] were screened from 23,182 samples referred for vancomycin concentration measurements from January 2018 to December 2019. Of the 3370 samples, we finally selected 28 patients with confirmed VAKI (VAKI group) by medical records review. VAKI was defined as follows: (1) an increase in SCr level of at least 0.5 mg/dL or (2) an increase from the pre-vancomycin treatment SCr of at least 50% from baseline on two consecutive days in the absence of an alternative explanation [22]. The control group consisted of three subgroups of 23 patients without VAKI who were suspected to have bacterial infections and who received vancomycin treatment (infection subgroup); 23 patients with CKD who were outpatients of the nephrology department with GFRs of less than 60 mL/min/1.73 m^2^ and not receiving vancomycin treatment (CKD subgroup); and 23 subjects referred for routine health examination without abnormal medical findings (healthy control (HC) subgroup). Each subgroup was matched by age and sex to the VAKI group. All subjects met the following criteria: (1) age of at least 18 years; (2) no antidepressant drug therapy, including selective serotonin reuptake inhibitors; (3) no hemodialysis; and (4) no current pregnancy.

The study was conducted according to the guidelines of the Declaration of Helsinki and was approved by the Institutional Review Board of Samsung Medical Center Hospital (IRB file no. 2017-12-038). A waiver of consent was obtained given the nature of the project. The study design and cohort subjects are described in Figure 1.

### 2.2. Blood Sampling

Clinical samples referred for SCr and vancomycin concentration from subjects with vancomycin treatment (VAKI group and infection subgroup) or referred for SCr from subjects without vancomycin treatment (CKD subgroup and HC subgroup) were collected, and residual serum samples were used for our metabolomics analyses. Peripheral venous blood samples were collected by venipuncture into a serum separator tube and centrifuged for 10 min at 2270× *g*, and the serum was separated and stored in an Eppendorf tube at −70 °C until analysis.

### 2.3. The Exploratory Metabolome Profiling Analysis

Among 97 subjects, exploratory metabolome profiling analysis was performed using seven serum samples, including from two subjects in the VAKI group, two subjects in the infection subgroup, and three subjects in the HC subgroup (Appendix A). To avoid interference with metabolomics profiling, we selectively included subjects without CKD history, previous malignancy, or chemotherapy by medical record review. The analysis was performed by Human Metabolome Technologies, Inc. (Yamagata, Japan) using capillary electrophoresis time-of-flight mass spectrometry with a fused silica capillary inner diameter of 50 μm × 80 cm (Agilent Technologies, Santa Clara, CA, USA) in two modes for cationic and anionic metabolites, as previously described [23]. Comparative analysis and hierarchical cluster analysis were performed using the relative peak area of putative metabolites. The Visualization and Analysis of Networks Containing Experimental Data (VANTED) software (VANTED version 2.5, www.vanted.org, accessed on 28 April 2021) was used for creating metabolic pathway maps based on peak profiles of putative metabolites.

### 2.4. The Amino Acids Profiling Analysis

Serum concentrations of 43 amino acids and amino acid derivatives were measured using the Agilent 6460 Triple Quadrupole MS/MS (Agilent Technologies Inc.), as previously described [24]. Briefly, protein precipitation was performed aTRAQ^TM^ kits for Amino Acid Analysis in Physiological Fluids (AB Sciex, Foster City, CA, USA) according to the manufacturer’s instructions. The Agilent 1260 Infinity LC system (Agilent Technologies Inc.) with a reverse-phase C18 (5 mm × 4.6 mm × 9150 mm) column at 50 °C was used to separate amino acids.

### 2.5. Measurement of 5-HT, and 5-HIAA

Serotonin (5-HT) and 5-hydroxyindoleacetic acid (5-HIAA) concentrations were measured by NeuroVIS (Cheonan, Korea), using AB SCIEX Triple Quadrupole 6500+ (AB Sciex, Framingham, MA, USA), in ExionLC™ Series UHPLC (AB Sciex, Framingham, MA, USA) with ACQUITY UPLC HSS T3 (2.1 × 100 mm, 1.8 µm, Waters, Milford, MA, USA) column at 30 °C was used to separate 5-HT and 5-HIAA by a gradient mobile phase of 0.1% formic acid in water and 5 mM of ammonium formate in methanol at a flow rate of 0.3 mL/min. These substances were monitored using positive electrospray ionization in multiple-reaction monitoring (MRM) mode (Q1 mass (*m*/*z*)/ Q3 mass (*m*/*z*); 177.1/160.1 5-HT; 192.0/146.1 5-HIAA, respectively).

### 2.6. Measurement of Serum Creatinine and Vancomycin Concentration

SCr was measured by an IDMS traceable Jaffe and enzymatic assay with Cobas c702 module (Roche Diagnostics Corp., Indianapolis, IN, USA). The GFR was calculated with the Cockcroft–Gault equation. Samples for measurement of vancomycin concentration were collected within one hour of regularly scheduled vancomycin infusion and were measured by the Cobas c702 analyzer using VANC3 (Roche Diagnostics GmbH, Indianapolis, IN, USA).

### 2.7. Statistical Analysis

Fisher’s exact test or the chi-square test was used for categorical data. The Kruskal–Wallis test or Mann–Whitney U test with Bonferroni correction was used for continuous data. For discriminating metabolite concentrations between the VAKI group and non-VAKI subgroups, receiver operating characteristic (ROC) curves were drawn, and area under curve (AUC) analyses were performed. Data analyses were conducted using The Statistical Package for the Social Sciences version 25.0 (IBM Corporation, Armonk, NY, USA). *p*-values < 0.05 were considered statistically significant.

## 3. Results

### 3.1. Subject Characteristics

The clinical characteristics of this study cohort are summarized in Table 1. The age and sex distribution of a total of 97 subjects (mean age ± standard deviation (SD): 60.2 ± 13.1 years; 39% female) was balanced between the VAKI group and non-VAKI group. The body mass index (BMI) of the VAKI group was lower than that of the non-VAKI group (21.8 ± 3.2 kg/m^2^ vs. 24.0 ± 3.2 kg/m^2^; *p* = 0.002), whereas the BMI of the infection subgroup was comparable to that of the VAKI group. The SCr and GFR of the VAKI group were significantly different from those of the non-VAKI group (1.60 ± 0.96 mg/dL vs. 1.05 ± 0.51 mg/dL; *p* < 0.001 and 45.9 ± 12.8 mL/min/1.73 m^2^ vs. 76.8 ± 32.4 mL/min/1.73 m^2^; *p* < 0.001), whereas the SCr and GFR of the CKD subgroup were comparable to those of the VAKI group. The C-reactive protein (CRP) level of the VAKI group was higher than that of the non-VAKI group (5.78 ± 8.74 mg/dL vs. 2.39 ± 4.88 mg/dL; *p* < 0.001), whereas the CRP level of the infection subgroup was comparable to that of the VAKI group. The serum vancomycin C_min_ was significantly higher in the VAKI group than in the infection subgroup (27.3 ± 5.3 μg/mL vs. 14.2 ± 2.6 μg/mL; *p* < 0.001).

### 3.2. The Results of the Exploratory Metabolome Profiling Analysis

Of a total of 221 detected peaks, 192 were matched on the basis of the standard library of Human Metabolome Technologies, Inc. (Yamagata, Japan) from the exploratory metabolome profiling analysis. Hierarchical cluster analysis was performed using the relative peak area of putative metabolites (Figure 2A), and the total metabolomics pathway map was plotted with the obtained metabolome and statistic data (Appendix A). In the pathway map, amino acid metabolism as well as central carbon metabolism, urea cycle, lipid metabolism, nucleotide metabolism, and metabolism of coenzymes showed differences between the two subjects from the VAKI group and the five subjects from the non-VAKI group. When sorted by Human Metabolome Database (HMDB) classification, the concentrations of indolic compounds such as 3-indoxylsulfuric acid, IAA, and 5-MIAA were greater in the two subjects from the VAKI group compared with the five subjects from the non-VAKI group, whereas 5-HT and tryptophan (Trp) were not (Figure 2B). Similarly, when sorted by VANTED pathway cluster categorization, the concentrations of Trp metabolism compounds such as 5-MIAA, anthranilic acid, IAA, kynurenine, and quinolinic acid were greater in the two subjects from the VAKI group compared with the five subjects from the non-VAKI group, whereas 5-HT and Trp were not (Figure 2B).

### 3.3. The Results of the Amino Acids Profiling Analysis

A total of 43 amino acids and amino acid derivatives were measured by the amino acids profiling analysis. The comparisons of these concentrations between the VAKI group and non-VAKI group were summarized in Table 2. Beta-alanine, anserine, gamma-aminobutyric acid, aspartic acid, ethanolamine, glutamic acid, methionine, O-phosphoserine, taurine, argininosuccinic acid, and all-isoleucine showed reduced serum concentrations in the VAKI group compared to non-VAKI group, whereas beta-aminoisobutyric acid, citrulline, 3-methylhistidine, and homocitrulline showed elevated serum concentrations in the VAKI group compared to the non-VAKI group, respectively. Among these compounds, gamma-aminobutyric acid and O-phosphoserine showed reduced serum concentrations in the VAKI group compared to all of the non-VAKI subgroups, whereas citrulline and homocitrulline showed significantly elevated serum concentrations in the VAKI group compared to all of the non-VAKI subgroups, respectively. Trp showed no significant difference between the VAKI group and the non-VAKI group, while Trp showed an elevated serum concentration in the VAKI group compared to the HC subgroup.

### 3.4. The Results of Measurement of 5-HT and 5-HIAA

The comparisons of the serum concentrations of Trp, 5-HT, 5-HIAA, and the ratios of 5-HT/Trp and 5-HIAA/5-HT between the VAKI group and the non-VAKI group are summarized in Table 3. Trp concentrations showed no significant difference between the VAKI group and the total non-VAKI group (median, 21.0 μmol/mL vs. 25.0 μmol/mL, *p* = not significant), while Trp showed a reduced concentration in the VAKI group compared with the HC subgroup (21.0 μmol/mL vs. 35.4 μmol/mL, *p* = 0.017) (Figure 3A). 5-HT and 5-HT/Trp showed a reduced concentration (47.8 ng/mL vs. 251.2 ng/mL, *p* < 0.001) (Figure 3B) and ratio (2.0 vs. 9.8, *p* < 0.001) (Figure 3D) in the VAKI group compared to the non-VAKI group, whereas 5HIAA and 5-HIAA/5-HT showed elevated concentrations (249.4 ng/mL vs. 84.5 ng/mL, *p* < 0.001) (Figure 3C) and ratio (6.3 vs. 0.4, *p* < 0.001) (Figure 3E). Among the non-VAKI subgroups, the 5-HT concentration of the infection subgroup was lower than that of the CKD subgroup (87.5 ng/mL vs. 337.6 ng/mL, *p* = 0.011) or HC subgroup (87.5 ng/mL vs. 345.2 ng/mL, *p* = 0.015) (Figure 3B), while the 5-HIAA/5-HT ratio in the infection subgroup was higher than that of HC subgroup (0.9 vs. 0.3, *p* = 0.016) (Appendix A and Figure 3E). The concentration of 5-HIAA showed no significant differences between the infection subgroup, CKD subgroup, and HC subgroup (Figure 3C).

### 3.5. The AUC-ROC Analysis of 5-HT 5-HIAA, and 5-HIAA/5-HT for VAKI

The AUC analysis of 5-HT, 5-HIAA, and 5-HIAA/5-HT for VAKI is illustrated in Figure 4. Of 97 subjects in the VAKI group and non-VAKI group, the AUCs for 5-HT, 5-HIAA, and 5-HIAA/5-HT were 0.200 (95% confidence interval, 0.105–0.296), 0.795 (0.704–0.885), and 0.884 (0.881–0.956), respectively (Figure 4A). Of 51 subjects of the VAKI group and infection subgroup, the AUCs for 5-HT, 5-HIAA, and 5-HIAA/5-HT were 0.345 (0.193–0.496), 0.747 (0.608–0.886), and 0.795 (0.671–0.919), respectively (Figure 4B). Of 51 subjects in the VAKI group and CKD subgroup, the AUCs for 5-HT, 5-HIAA, and 5-HIAA/5-HT were 0.149 (0.039–0.259), 0.848 (0.744–0.951), and 0.925 (0.852–0.999), respectively (Figure 4C). Finally, of 51 subjects in the VAKI group and HC subgroup, the AUCs for 5-HT, 5-HIAA, and 5-HIAA/5-HT were 0.107 (0.019–0.195), 0.789 (0.664–0.914), and 0.930 (0.866–0.995), respectively (Figure 4D).

## 4. Discussion

Our study is the first comparative metabolomics investigation of serum concentrations of Trp and its metabolites 5-HT, 5-HIAA, and 5-HIAA/5-HT in VAKI and non-VAKI groups, with the latter group consisting of the infection, CKD, and HC subgroups. Trp is an essential amino acid in human nutrition and is a substrate for tissue protein synthesis [25,26]. In total, 95% of Trp is metabolized via the kynurenine pathway [27], and the ratio of kynurenine to Trp (Kyn/Trp) has been studied for its impact in the settings of inflammation, neurodegeneration, and end-stage kidney disease (ESRD) [28,29,30,31]. Other metabolites of the kynurenine pathway, such as anthranilic acid and indole-3-acetic acid (IAA), have been reported as possible surrogate markers for inflammation and ESRD [32,33]. The results from our exploratory metabolome profiling analysis revealed that VAKI patients had higher serum values of 5-methoxyindoleacetic acid (5-MIAA), anthranilic acid, IAA, kynurenine, quinolinic acid, and Kyn/Trp compared with non-VAKI patients (Figure 2B), and these findings were similar to those of a previous report in AKI patients [34]. However, Kyn/Trp might not be suitable for detecting VAKI because it tends to be higher in patients with infection and CKD compared with HCs [32,33,35,36]. Furthermore, Kyn/Trp and other metabolites of the kynurenine pathway would be affected by dietary conditions, including Trp supplementation [37].

Interestingly, our data revealed a gradual increase of the 5-HT pathway components 5-HT and 5-MIAA in VAKI subjects compared with non-VAKI subjects, while 5-HIAA was not detected in the exploratory metabolome profiling analysis (Figure 2B). 5-HT pathway compounds, such as 5-HT, 5-HIAA, and 5-MIAA, are derived from 5% of Trp [27]. 5-HT is synthesized by tryptophan hydroxylase and aromatic L-amino acid decarboxylase via 5-hydroxytryptophan [38]. 5-HIAA is a primary metabolite derived from 5-HT by mitochondrial monoamine oxidase A (MAO-A) [39], and the ratio of 5-HIAA to 5-HT (5-HIAA/5-HT) is a surrogate marker for serotonergic activity or MAO-A activity [40,41]. MAO-A is an enzyme that generates hydrogen peroxide, one of the sources of oxygen free radicals [42]. An increase in oxygen free radicals is attributed to oxidative stress in living organisms. Proinflammatory oxidation, mitochondrial dysfunction, and cellular apoptosis have been studied as principal mechanisms of VAKI [6]. Both 5-HT and 5-HIAA have been considered in the occurrence of ischemia/reperfusion damage in experimental animal models [43,44]. Khovanskaya et al. reported an increase in urinary 5-HIAA concentration, consistent with a period of decrease in tissue 5-HT level, in an ischemic kidney model of rabbits [43]. Du et al. showed that increase in interstitial 5-HIAA and decrease in interstitial 5-HT occurred at a reperfusion stage rather than an ischemia stage in an ischemic myocardial rat model [44]. These phenomena are explained by the role of serotonin as an antioxidant [45] and increased MAO-A activity in oxidative stress condition [46,47]. Therefore, we focused on 5-HT and 5-HIAA as potential biomarkers for detecting VAKI among candidate compounds from exploratory metabolome profiling analysis.

BCAAs and sulfur-containing amino acids (SCAAs) are used for nutrient supplement therapy in septic patients or in liver disease [48,49], and these amino acids are considered as antioxidants to protect the oxidative stress [48,49,50]. Carnosine associated amino acids (CAAs) act a role as a physiological buffer in human [51,52], and the previous study has been examined with protective effects of CAAs in AKI due to septic shock in a rat model [52]. In the results from our amino acids profiling analysis, BCAAs (such as all-isoleucine, leucine and valine) and SCAAs (such as serine, O-phosphoserine, methionine, and taurine) showed reduced concentrations in the VAKI group and CKD subgroup compared to the HC subgroup, whereas urea cycle associated amino acids (such as arginine, citrulline, and homocitrulline), and cystine showed elevated concentrations in the VAKI group compared to the CKD subgroup, respectively (Table 2 and Appendix A). Furthermore, CAAs (such as anserine and carnosine) showed reduced concentrations in the VAKI group and CKD subgroup compared to the HC subgroup, whereas their precursors, such as beta-alanine and 1-methyhistidine, showed elevated concentrations in the infection subgroup compared to the VAKI group (Table 2 and Appendix A). These results suggest that the reduced concentrations of Trp and 5-HT were associated with oxidative stress in the VAKI group.

AKI biomarker concentrations also were high in patients with CKD, and quantitative markers and cutoff thresholds would be useful for distinguishing biomarker elevations in VAKI patients with CKD [13]. In the current study, the 5-HIAA concentration was three-fold higher and the 5-HIAA/5-HT ratio was 15-fold higher in the VAKI group than in the non-VAKI group, whereas the 5-HT concentration was one-fifth lower in the VAKI group than in the non-VAKI group (Figure 3D). The AUCs for 5-HT and 5-HIAA were comparable between the VAKI group and non-VAKI group. Meanwhile, the AUC for the 5-HIAA/5-HT ratio was superior for identifying VAKI compared with those of 5-HT and 5-HIAA (Figure 4A). In the VAKI group and the CKD subgroup or HC subgroup, AUC-ROC for 5-HIAA/5-HT were 0.925 or 0.930, respectively (Figure 4C,D). However, when contrasting the VAKI group and the infection subgroup, the AUC for 5-HIAA/5-HT was 0.795, which was the lowest AUC value recorded among the non-VAKI subgroups (Figure 4B). When the cut-off value of 5-HIAA/5-HT ratio was set to 2.0 using the maximum Kolmogorov–Smirnov metric, specificity was lowest between the VAKI group and the infection subgroup (VAKI vs. infection, 78.3%; VAKI vs. CKD, 91.3%; VAKI vs. HC, 100%), while the sensitivity was the same at 75.0% for all non-VAKI subgroups. Our data showed that the 5-HIAA/5-HT ratios of the infection subgroup and the VAKI group were higher than that of the HC subgroup (Appendix A and Figure 3E), and these results were similar to those of a previous study involving septic shock patients [34]. However, the CKD subgroup displayed relatively increased 5-HT and decreased Trp concentrations compared with the other groups, and the 5-HT/Trp ratio of the CKD subgroup was significantly higher than that of the VAKI group and the infection subgroup. Depending on the decrease in GFR, the 5-HT/Trp or 5-HIAA/5-HT ratio showed different alteration in the CKD subgroup or VAKI group (Figure 5A,B). These findings suggest that the 5-HIAA/5-HT ratio is a surrogate marker for acute reactive reactions such as inflammation or hypoxia rather than chronic kidney damage or dysfunction, while the 5-HT/Trp ratio can indicate chronic kidney damage or dysfunction in patients with CKD.

Furthermore, our study revealed that the serum 5-HT and 5-HIAA concentrations were not related to serum Trp concentration (Table 3 and Figure 3A,B), whereas the serum kynurenine concentration and Kyn/Trp ratio were affected by dietary Trp supplementation or serum Trp concentration [37,53]. These findings indicate that 5-HT and 5-HIAA would be more stable markers than serum kynurenine and Kyn/Trp despite nutrient intake and sociodemographic factors.

Our study has limitations. We collected single serum samples referred for Scr from subjects with elevated vancomycin C_min_. An additional study with serial samples from the initial period of vancomycin treatment should be conducted to evaluate the performance of the 5-HIAA/5-HT ratio for earlier detection of VAKI.

Taken together, our results reveal that the serum concentration of 5-HT was independently decreased and the serum concentration of 5-HIAA and the 5-HIAA/5-HT ratio were increased in the VAKI group compared with all non-VAKI subgroups regardless of the serum Trp concentration.

## 5. Conclusions

In conclusion, our study demonstrated that increased 5-HIAA/5-HT ratio has potential to act as a novel biomarker for VAKI detection. Additionally, we showed that the 5-HIAA/5-HT ratio would be a useful indicator for oxidative stress and inflammation.

## Figures and Tables

**Figure 1 antioxidants-10-00895-f001:**
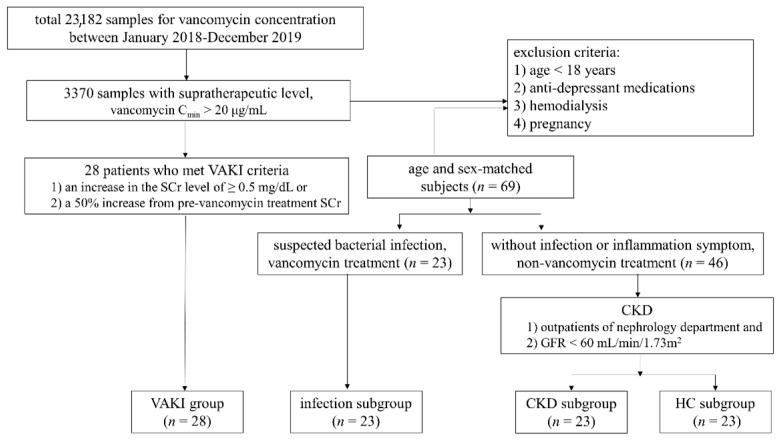
Schematic study flow and cohort population. Abbreviations: VAKI, vancomycin-associated acute kidney injury; CKD, chronic kidney disease; HC, healthy control; C_min_, trough concentration; GFR, glomerular filtration rate; SCr, serum creatinine concentration.

**Figure 2 antioxidants-10-00895-f002:**
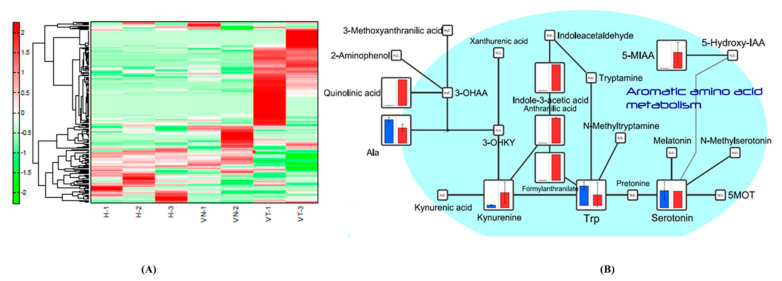
Serotonin pathway map and kynurenine pathway map with the obtained metabolome data from exploratory metabolome profiling analysis (cropped). Notes: Hierarchical cluster analysis was performed using the relative peak area of putative metabolites from seven serum samples, including from two subjects in the VAKI group (VT-1 and VT-3), two subjects in the infection subgroup (VN-1 and VN-2), and three HCs (H-1, H-2, and H-3) (**A**), and the metabolomics pathway map was created using the VANTED software (**B**). The blue and red boxes represent the mean relative peak area of the component in non-VAKI subjects and VAKI subjects, respectively. The cropped serotonin pathway map and kynurenine pathway map showed that VAKI subjects had significantly increased concentrations of indolic compounds such as 3-indoxylsulfuric acid, IAA, and 5-MIAA compared with non-VAKI subjects. Abbreviations: 3-OHAA, 3-hydroxyanthranilic acid; 3-OHKY, 3-hydroxykynurenine; 5-MOT, 5- methoxytryptamine; 5-hydroxy-IAA, 5-hydroxyindoleacetic acid; 5-MIAA, 5-methoxyindole-3-acetic acid; Ala, alanine; Trp, tryptophan; VANTED, Visualization and Analysis of Networks Containing Experimental Data.

**Figure 3 antioxidants-10-00895-f003:**
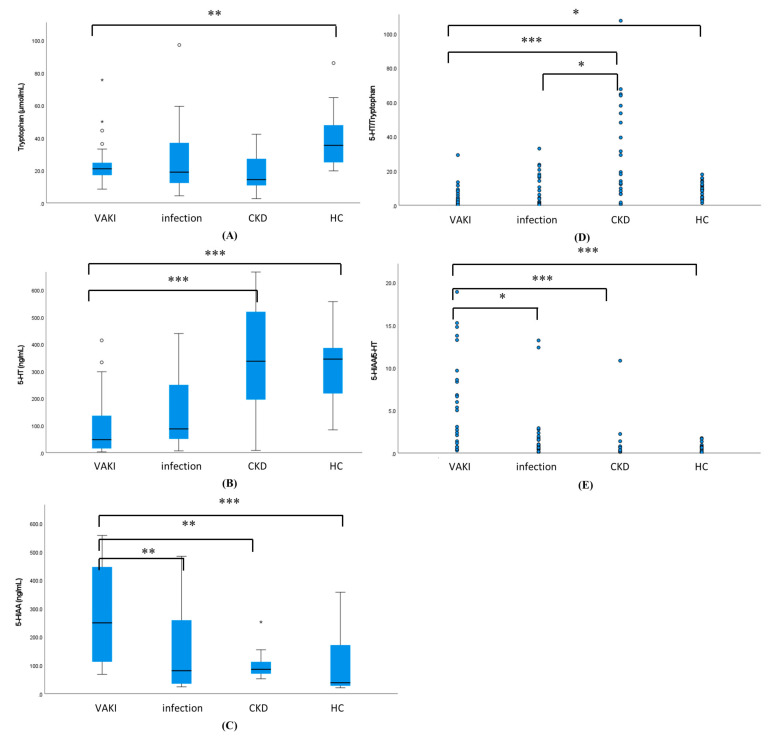
Comparison of serum concentration of tryptophan, 5-HT, 5-HIAA, and the ratio of 5-HT to tryptophan (5-HT/Trp) and 5-HIAA to 5-HT (5-HIAA/5-HT) between the VAKI group and non-VAKI subgroup. Notes: (**A**) Serum concentration of tryptophan (μmol/mL); (**B**) Serum concentration of 5-HT (ng/mL); (**C**) Serum concentration of 5-HIAA (ng/mL); (**D**) the ratio of 5-HT to tryptophan (5-HT/Trp); and (**E**) the ratio of 5-HIAA to 5-HT (5-HIAA/5-HT) were compared between the VAKI group and non-VAKI subgroups. Each box represents the interquartile range. Each line inside the boxes is plotted at the median. Each whisker means the 10th and 90th percentiles. * *p* < 0.5, ** *p* < 0.01, *** *p* < 0.001, respectively. Abbreviations: VAKI, vancomycin-associated acute kidney injury; CKD, chronic kidney disease; HC, healthy control; 5-HT, serotonin; 5-HIAA, 5-hydroxyindoleacetic acid; 5-HT/Trp, the ratio of serotonin to tryptophan; 5-HIAA/5-HT, the ratio of 5-hydroxyindoleacetic acid to serotonin.

**Figure 4 antioxidants-10-00895-f004:**
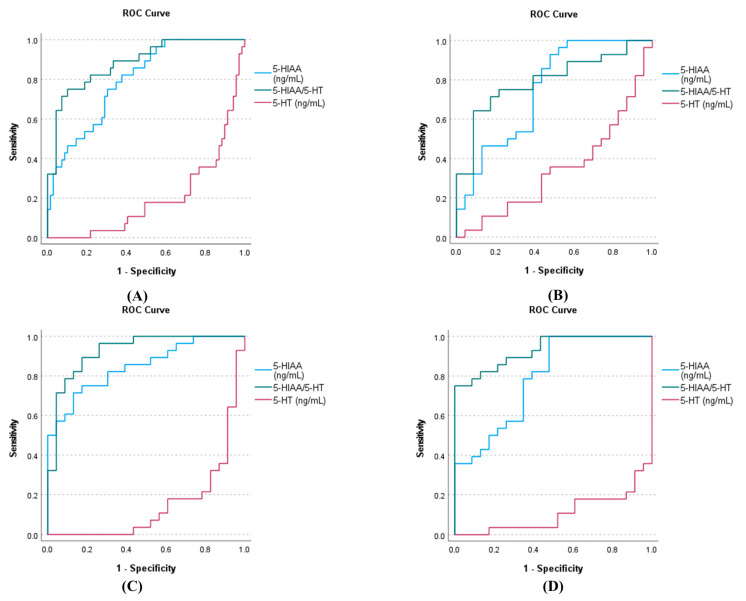
AUC analysis of 5-HT, 5-HIAA, and 5-HIAA/5-HT for VAKI in the VAKI group and non-VAKI subgroups. Notes: The AUC analysis of 5-HT, 5-HIAA, and 5-HIAA/5-HT was performed (**A**) for the VAKI group and all of the non-VAKI subgroups, (**B**) for the VAKI group and infection subgroup, (**C**) for the VAKI group and CKD subgroup, and (**D**) for the VAKI group and HC subgroup. Abbreviations: VAKI, vancomycin associated acute kidney injury; CKD, chronic kidney disease; HC, healthy control; 5-HT, serotonin; 5-HIAA, 5-hydroxyindoleacetic acid; 5-HT/Trp, the ratio of 5-hydroxyindoleacetic acid to serotonin.

**Figure 5 antioxidants-10-00895-f005:**
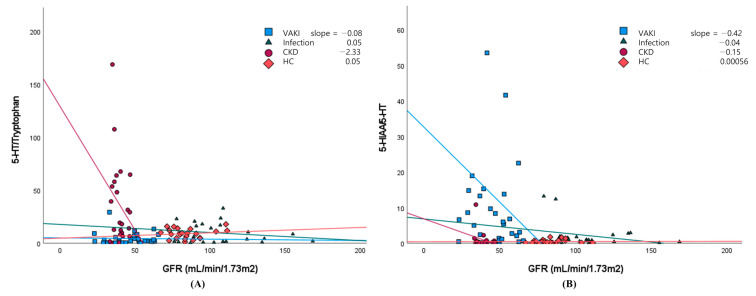
Scatterplots assessing the relationship between GFR and the 5-HT/Trp or 5-HIAA/5-HT ratio in the VAKI group and non-VAKI subgroups. Notes: The scatterplots assessing the relationship between GFR (mL/min/1.73 m^2^) and the 5-HT/Trp ratio (**A**) or 5-HIAA/5-HT ratio (**B**) were plotted in the VAKI group, infection subgroup, CKD subgroup, and HC subgroup. The reference line of the 5-HT/Trp ratio showed a markedly negative slope for GFR in the CKD subgroup (purple), whereas that of the 5-HIAA/5-HT ratio showed a markedly negative slope for GFR in the VAKI group (blue). Abbreviations: VAKI, vancomycin-associated acute kidney injury; CKD, chronic kidney disease; HC, healthy control; GFR, glomerular filtration rate; SCr, serum creatinine concentration; Trp, tryptophan; 5-HT, serotonin; 5-HIAA, 5-hydroxyindoleacetic acid; 5-HT/Trp, the ratio of serotonin to tryptophan; 5-HIAA/5-HT, the ratio of 5-hydroxyindoleacetic acid to serotonin.

**Table 1 antioxidants-10-00895-t001:** Clinical characteristics of subjects in this study.

Clinical Characteristics	Overall	VAKI	Non-VAKI							
			Total Non-VAKI	*p*-Value ^a^	Infection	*p*-Value ^a^	CKD	*p*-Value ^a^	HC	*p*-Value ^a^
Number of Subjects	97	28	69		23		23		23	
Age, yr (mean ± SD)	60.2 ± 13.1	60.4 ± 14.5	60.2 ± 13.1	N.S	58.4 ± 12.2	N.S	63.4 ± 12.1	N.S	58.4 ± 13.1	N.S
range	20–85	20–85	21–82		37–82		21–72		24–81	
Female (%)	38 (39%)	10 (36%)	28 (41%)	N.S	9 (39%)	N.S	9 (39%)	N.S	10 (43%)	N.S
BMI (kg/m^2^)	23.4 ± 3.4	21.8 ± 3.2	24.0 ± 3.2	0.002	23.2 ± 3.5	N.S	24.3 ± 3.4	0.018	24.6 ± 2.7	0.02
SCr (mg/dL)	1.16 ± 0.60	1.60 ± 0.96	1.05 ± 0.51	<0.001	0.62 ± 0.21	<0.001	1.61 ± 0.29	N.S	0.85 ± 0.15	<0.001
GFR (mL/min/1.73 m^2^)	67.8 ± 31.4	45.9 ± 12.8	76.8 ± 32.4	<0.001	107.8 ± 23.9	<0.001	39.5 ± 4.3	N.S	84.6 ± 11.8	<0.001
CRP (mg/dL)	0.26 ± 6.78	5.78 ± 8.74	2.39 ± 4.88	<0.001	7.19 ± 6.31	N.S	0.07 ± 0.94	<0.001	0.04 ± 0.71	<0.001
Vancomycin C_min_		27.3 ± 5.3			14.2 ± 2.6	<0.001				
(μg/mL)										

Notes: a Each *p*-value was calculated relative to the VAKI group. Abbreviations: VAKI, vancomycin-associated acute kidney injury; CKD, chronic kidney disease; HC, healthy control; SD, standard deviation; BMI, body mass index; SCr, serum creatinine; CRP, C reactive protein; C_min_, trough concentration.

**Table 2 antioxidants-10-00895-t002:** Comparison of serum amino acids and amino acid derivatives concentration between VAKI group and non-VAKI group.

Parameters	VAKI	Non-VAKI							
(μmol/mL)		Total Non-VAKI	*p*-Value ^a^	Infection	*p*-Value ^a^	CKD	*p*-Value ^a^	HC	*p*-Value ^a^
Alanine	417.3 (324.0–487.3)	296.0 (207.5–393.3)	0.002	290.6 (241.5–529.9)	N.S	201.2 (141.7–267.2)	<0.001	371.7 (339.4–420.3)	N.S
Beta-alanine	2.1 (1.4–2.9)	4.7 (2.4–7.2)	<0.001	7.3 (6.2–8.6)	<0.001	3.5 (2.4–6.5)	0.01	2.4 (1.9–4.7)	N.S
Anserine	0.0 (0.0–0.0)	0.0 (0.0–0.7)	<0.001	0.5 (0.0–0.8)	<0.001	0.0 (0.0–0.0)	N.S	0.6 (0.0–0.9)	<0.001
Aminoadipic acid	0.0 (0.0–1.0)	0.6 (0.0–1.1)	N.S	0.6 (0.0–1.0)	N.S	0.7 (0.0–1.1)	N.S	0.5 (0.0.–0.9)	N.S
Alpha-aminobutyric acid	10.2 (8.2–13.4)	13.5 (8.5–17.9)	0.05	14.9 (11.4–19.1)	0.002	6.4 (3.9–11.5)	0.037	16.6 (11.7–20.0)	<0.001
Gamma-aminobutyric acid	0.0 (0.0–0.2)	1.4 (0.9–2.0)	<0.001	1.9 (1.3–2.5)	<0.001	0.9 (0.5–1.9)	<0.001	1.3 (1.1–1.6)	<0.001
Beta-aminoisobutyric acid	2.8 (1.4–4.8)	0.7 (0.0–2.3)	0.002	0.5 (0.0–1.7)	0.001	2.6 (0.0–9.8)	N.S	0.7 (0.0–1.1)	<0.001
Arginine	72.8 (59.2–84.2)	61.9 (37.3–95.0)	N.S	62.0 (40.6–143.2)	N.S	37.3 (23.2–59.1)	<0.001	73.7 (59.4–117.9)	N.S
Asparagine	29.5 (23.7–34.9)	28.2 (19.4–36.3)	N.S	30.1 (20.4–43.0)	N.S	19.4 (13.0–33.5)	0.011	30.5 (23.2–36.5)	N.S
Aspartic acid	11.3 (8.5–16.4)	19.6 (11.3–36.0)	0.01	24.2 (19.0–41.6)	<0.001	11.8 (9.1–37.3)	N.S	15.3 (10.4–32.6)	N.S
Carnosine	0.0 (0.0–0.0)	0.0 (0.0–0.0)	0.024	0.0 (0.0–0.3)	0.017	0.0 (0.0–0.0)	N.S	0.0 (0.0–0.0)	0.039
Citrulline	35.0 (29.3–39.4)	20.9 (13.2–30.5)	<0.001	17.6 (11.2–39.5)	0.002	14.1 (9.8–22.0)	<0.001	27.4 (22.4–33.0)	0.007
Cystathionine	0.0 (0.0–0.0)	0.0 (0.0–0.8)	N.S	0.0 (0.0–0.0)	N.S	0.8 (0.0–1.6)	0.002	0.6 (0.0–0.5)	N.S
Cystine	3.7 (1.9–8.1)	5.6 (2.0–13.0)	N.S	11.4 (7.8–13.5)	<0.001	10.4 (5.2–29.4)	<0.001	1.2 (0.0–2.0)	<0.001
Ethanolamine	5.8 (4.8–6.6)	9.1 (5.5–11.9)	0.037	8.9 (7.0–11.6)	0.01	5.5 (3.4–10.3)	N.S	10.9 (6.7–15.4)	0.003
Glutamic acid	54.1 (39.8–76.9)	82.2 (54.1–114.5)	0.016	100.4 (72.5–158.6)	0.002	58.9 (35.6–113.1)	N.S	72.0 (56.4–99.6)	0.034
Glutamine	313	257.5	N.S	232.6	0.022	140.5	<0.001	382.3	0.028
	(275.1–370.9)	(145.1–378.5)		(170.5–326.9)		(97.4–244.7)		(298.5–570.1)	
Glycine	136	142.7	N.S	149.7	N.S	110.3	0.043	165.9	N.S
	(117.2–198.9)	(119.8–266.4)		(125.4–267.9)		(98.4–244.7)		(133.3–288.5)	
Histidine	34.9 (32.1–38.6)	40.2 (29.4–57.7)	N.S	37.1 (28.2–54.3)	N.S	35.7 (17.3–46.6)	N.S	47.0 (37.7–80.2)	0.001
Homocysteine	0.0 (0.0–0.0)	0.0 (0.0–0.0)	N.S	0.0 (0.0–0.0)	N.S	0.0 (0.0–0.0)	N.S	0.0 (0.0–0.0)	N.S
Dehydroxylysine	0.0 (0.0–0.0)	0.0 (0.0–0.0)	N.S	0.0 (0.0–0.0)	N.S	0.0 (0.0–0.0)	N.S	0.0 (0.0–0.0)	N.S
Hydroxyproline	7.6 (5.4–10.4)	6.1 (3.9–9.4)	N.S	6.6 (3.9–9.4)	N.S	6.1 (3.1–14.9)	N.S	5.2 (4.4–8.8)	N.S
Isoleucine)	32.0 (25.8–39.0)	35.2 (25.5–51.5)	N.S	41.0 (31.1–59.0)	0.041	24.4 (16.4–40.7)	N.S	37.9 (29.2–53.1)	N.S
Leucine	80.2 (59.3–99.1)	93.8 (72.1–137.3)	N.S	103.1 (87.4–140.8)	0.015	53.1 (34.5–105.9)	N.S	106.7 (81.6–147.4)	0.006
Lysine	99.2 (83.1–114.8)	122.2 (78.2–157.5)	N.S	122.2 (83.7–170.2)	N.S	73.1 (46.0–127.1)	N.S	131.1	0.001
								(105.5–185.6)	
Methionine	6.0 (5.1–8.9)	16.1 (7.2–29.7)	0.002	9.3 (5.9–15.9)	N.S	8.0 (4.6–15.9)	N.S	34.7 (26.9–40.0)	<0.001
1-methylhistidine	7.4 (5.9–9.9)	9.2 (6.2–12.6)	N.S	13.6 (10.1–14.9)	<0.001	6.3 (4.1–10.1)	N.S	8.8 (6.2–10.3)	N.S
3-methylhistidine	4.8 (4.0–7.2)	3.4 (1.6–6.8)	0.024	1.7 (0.9–10.7)	0.018	5.7 (4.2–12.3)	N.S	2.1 (1.6–4.3)	<0.001
Ornithine	48.9 (38.5–77.1)	53.7 (40.9–92.6)	N.S	88.4 (51.3–117.3)	0.039	51.9 (35.7–90.3)	N.S	49.8 (42.5–69.9)	N.S
Phenylalanine	51.5 (46.2–67.7)	64.6 (47.0–87.6)	N.S	76.4 (55.5–100.3)	0.003	65.4 (40.9–104.7)	N.S	54.9 (37.2–82.4)	N.S
O-phosphoethanolamine	0.0 (0.0–0.0)	0.0 (0.0–0.0)	N.S	0.0 (0.0–0.0)	N.S	0.0 (0.0–0.0)	N.S	0.0 (0.0–0.0)	N.S
O-phosphoserine	1.6 (1.2–2.0)	3.0 (1.7–4.6)	<0.001	3.8 (1.9–.5.1)	0.002	2.4 (1.8–3.5)	0.009	3.4 (1.6–8.3)	0.002
Proline	97.7 (76.9–127.9)	83.3 (49.3–129.9)	N.S	89.5 (86.7–141.9)	N.S	53.2 (44.4–106.1)	0.013	74.9 (57.1–169.0)	N.S
Sarcosine	0.0 (0.0–0.6)	0.6 (0.0–0.8)	N.S	0.6 (0.0–1.2)	N.S	0.0 (0.0–0.8)	N.S	0.6 (0.0–0.8)	N.S
Serine	78.2 (69.2–93.9)	88.0 (70.1–138.7)	N.S	100.3 (86.7–141.9)	0.01	63.3 (40.8–82.7)	0.021	111.0 (75.1–1843.4)	0.009
Taurine	40.6 (32.6–50.4)	65.2 (34.7–99.3)	0.02	69.8 (39.9–109.3)	N.S	34.7 (29.1–73.9)	N.S	81.0 (60.8–127.1)	<0.001
Threonine	66.2 (52.9–80.8)	59.0 (40.7–89.4)	N.S	64.6 (46.0–114.2)	N.S	38.1 (30.4–63.9)	0.001	59.8 (50.3–118.3)	N.S
Tryptophan	21.0 (17.6–24.6)	25.0 (13.0–36.9)	N.S	18.9 (12.2–37.0)	N.S	14.3 (10.8–27.2)	N.S	35.4 (25.0–47.8)	<0.001
Tyrosine	34.1 (26.7–37.2)	39.1 (26.9–57.9)	N.S	36.4 (25.2–37.0)	N.S	37.2 (18.4–54.4)	N.S	41.7 (25.0–48.6)	0.011
Valine	129.3	142.1	N.S	178.1	0.034	91.9 (51.4–140.8)	0.009	156.9	0.014
	(114.9–160.5)	(101.3–220.3)		(138.0–222.2)				(125.5–256.8)	
Argininosuccinic acid	0.0 (0.0–0.0)	0.0 (0.0–1.1)	<0.001	0.6 (0.0–1.3)	<0.001	0.0 (0.0–0.6)	0.002	0.0 (0.0–2.0)	<0.001
Homocitrulline	1.1 (0.7–1.5)	0.0 (0.0–0.7)	<0.001	0.0 (0.0–0.4)	<0.001	0.0 (0.0–0.8)	0.001	0.0 (0.0–0.7)	<0.001
All-isoleucine	0.0 (0.0–0.0)	0.0 (0.0–0.8)	0.003	0.0 (0.0–0.0)	N.S	0.0 (0.0–0.9)	0.007	0.8 (0.0–1.4)	<0.001

Notes: a Each *p* value was calculated using Mann–Whitey U test, compared to VAKI group. Abbreviations: VAKI, vancomycin associated acute kidney injury; CKD, chronic kidney disease; HC, healthy control.

**Table 3 antioxidants-10-00895-t003:** Comparison of serum tryptophan, serotonin, and 5-HIAA concentration between the VAKI group and non-VAKI groups.

Parameters	VAKI	Non-VAKI							
		Total Non-VAKI	*p*-Value ^a^	Infection	*p*-Value ^a^	CKD	*p*-Value ^a^	HC	*p*-Value ^a^
Trp (μmol/mL)	21.0 (17.6–24.6)	25.0 (13.0–36.9)	N.S	18.9 (12.2–37.0)	N.S	14.3 (10.8–27.2)	N.S	35.4 (25.0–47.8)	0.004
5-HT (ng/mL)	47.8 (16.6–135.5)	251.2 (114.5–389.3)	<0.001	87.5 (50.6–250.2)	N.S	337.6 (195.8–520.0)	<0.001	345.2 (218.4–386.4)	<0.001
5-HIAA (ng/mL)	249.4 (113.0–442.2)	84.5(36.3–154.5)	<0.001	80.8 (35.2–258.7)	0.006	85.7 (70.4–111.8)	0.003	38.5 (27.7–171.2)	<0.001
5-HT/Trp	2.0 (0.7–6.5)	9.8(3.8–18.0)	<0.001	4.3 (1.6–16.2)	N.S	20.0 (8.6–56.9)	<0.001	9.4 (4.8–12.1)	0.017
5-HIAA/5-HT	6.3 (1.9–14.0)	0.4 ( 0.2–0.9)	<0.001	0.9 (0.5–1.9)	0.032	0.2 (0.2–0.5)	<0.001	0.3 (0.1–0.7)	<0.001

Notes: a Each *p*-value was calculated using the Mann–Whitey U test and adjusted by Bonferroni correction relative to the VAKI group. Abbreviations: VAKI, vancomycin-associated acute kidney injury; CKD, chronic kidney disease; HC, healthy control; Trp, tryptophan; 5-HT, serotonin; 5-HIAA, 5-hydroxyindoleacetic acid; the ratio of serotonin to tryptophan; 5-HIAA/5-HT, the ratio of 5-hydroxyindoleacetic acid to serotonin.

## Data Availability

Not applicable.

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
