# Peer review of "Serum 5-Hydroxyindoleacetic Acid and Ratio of 5-Hydroxyindoleacetic Acid to Serotonin as Metabolomics Indicators for Acute Oxidative Stress and Inflammation in Vancomycin-Associated Acute Kidney Injury"

_antioxidants, 2021, doi:10.3390/antiox10060895_

Round 1

Reviewer 1 Report

The comments I have center around mechanisms of toxicity.

Can the author's explain in more detail the mechanist of nephrotoxicity?  What are possible interventions to potentially prevent or ameliorate deteriorating kidney function?

Author Response

Point 1: The comments I have center around mechanisms of toxicity.

Can the author's explain in more detail the mechanist of nephrotoxicity?  What are possible interventions to potentially prevent or ameliorate deteriorating kidney function?

Response 1: Thank you for the important suggestion. We explained the mechanism of nephrotoxicity and the possible interventions to potentially prevent loss of kidney function as follows:

“As a subtype of AKI, vancomycin-associated AKI (VAKI) is characterized by dose-dependent acute tubular necrosis, which is associated with oxidative stress in proximal tubule cells. VAKI is correlated closely with elderly, compromised renal function, concomitant nephrotoxic therapy, longer duration of vancomycin administration, high trough level of vancomycin, and critically ill status in the intensive care unit. The incidence of VAKI has been reported to range from 5% to 43%, and therapeutic drug monitoring and customized dosing of vancomycin have been studied for prevention of VAKI in a high-risk population.”

We have made these comments in “Introduction,” paragraph 1.

We thank Reviewer 1 for the comment.

Reviewer 2 Report

Lee et al. performed a comparative metabolomics search for novel biomarkers for vancomycin-associated acute kidney injury (VAKI), a complication of highest clinical relevance, which usually afflicts already critically ill patients and massively narrows their prognosis. They conclusively identified characteristic quantitative changes in several tryptophan metabolites in VAKI patients (n =28) compared to vancomycin-treated patients without VAKI (n=23) as well as patients with chronic kidney disease (n=23) and healthy controls (n=23). Especially an increased serum 5-hydroxyindolaceticacid to serotonin (5-HIAA/5-HT) ratio was able to discriminate between the first two clinically relevant groups (VAKI vs. non-VAKI vancomycin-treated patients) with a promising AUROC of 0.795.

The study is very well performed and presented and should definitely be published in “Antioxidants” after minor revisions:

  • The cut-off levels for which the presented AUROCs apply should be provided for the various parameters, especially for the most promising and easiest one to implement in clinical routine diagnostics, the 5-HIAA/5-HT ratio.
  • The time intervals between the blood draws of the samples utilized for the 5-HT and 5-HIAA and the blood draws from which the diagnosis of VAKI was made, based on serum creatinine levels, should be clearly stated. Where those earlier samples? The same samples? Is there any evidence form the present data, that the 5-HIAA/5-HT ratio allows for the earlier diagnosis of VAKI compared to serum creatinine? This is a critical issue as appropriately stated by the authors in the introduction and should also be addressed in the discussion.
  • Minor English language editing.

Author Response

Response to Reviewer 2 Comments

Lee et al. performed a comparative metabolomics search for novel biomarkers for vancomycin-associated acute kidney injury (VAKI), a complication of highest clinical relevance, which usually afflicts already critically ill patients and massively narrows their prognosis. They conclusively identified characteristic quantitative changes in several tryptophan metabolites in VAKI patients (n =28) compared to vancomycin-treated patients without VAKI (n=23) as well as patients with chronic kidney disease (n=23) and healthy controls (n=23). Especially an increased serum 5-hydroxyindolaceticacid to serotonin (5-HIAA/5-HT) ratio was able to discriminate between the first two clinically relevant groups (VAKI vs. non-VAKI vancomycin-treated patients) with a promising AUROC of 0.795.

The study is very well performed and presented and should definitely be published in “Antioxidants” after minor revisions:

Point 1: The cut-off levels for which the presented AUROCs apply should be provided for the various parameters, especially for the most promising and easiest one to implement in clinical routine diagnostics, the 5-HIAA/5-HT ratio.

Response 1: Thank you for the suggestion. We provided the cut-off level for the 5-HIAA/5-HT ratio for detecting VAKI using the maximum Kolmogorov-Smirnov metric, as follows:

“When the cut-off value of 5-HIAA/5-HT ratio was set to 2.0 using maximum Kolmogorov-Smirnov metric, the specificity was lowest between the VAKI group and infection subgroup (VAKI vs. infection, 78.3%; VAKI vs. CKD, 91.3%; VAKI vs. HC, 100%), while the sensitivity was the same at 75.0% for all non-VAKI subgroups.”

We have made these comments in “Discussion,” paragraph 4.

Point 2: The time intervals between the blood draws of the samples utilized for the 5-HT and 5-HIAA and the blood draws from which the diagnosis of VAKI was made, based on serum creatinine levels, should be clearly stated. Where those earlier samples? The same samples?

Response 2: Thank you for the question. We measured the 5-HIAA/5-HT ratio with the same sample used for measurement of serum creatinine. We clearly stated these sample data in “Discussion,” paragraph 6.

Point 3: Is there any evidence form the present data, that the 5-HIAA/5-HT ratio allows for the earlier diagnosis of VAKI compared to serum creatinine? This is a critical issue as appropriately stated by the authors in the introduction and should also be addressed in the discussion.

Response 3: Thank you for the important comments. These samples were collected from VAKI subjects with elevated vancomycin Cmin. We have specified the limitations of the study as follows.

“Our study has several limitations. We collected single serum samples referred for Scr from subjects with elevated vancomycin Cmin. Additional study with serial samples from the initial period of vancomycin treatment should be conducted to evaluate performance of the 5-HIAA/5-HT ratio for earlier detection of VAKI.”

We have made these comments in “Discussion,” paragraph 6.

Point 4: Minor English language editing.

Response 4: Thank you for the comment. The English in this revised manuscript has been assessed by native speakers of English.

We thank Reviewer 2 for the comments and suggestions.